# Comprehensive characterization of the effect of mineralocorticoid receptor antagonism with spironolactone on the renin-angiotensin-aldosterone system in healthy dogs

Allison K. Masters[1][¤]*, Jessica L. Ward[1], Emilie Guillot[2], Oliver Domenig[3], Lingnan Yuan[4], Jonathan P. Mochel[4]

**1** Department of Veterinary Clinical Sciences, College of Veterinary Medicine, Iowa State University, Ames, Iowa, United States of America, **2** Ceva Santé Animale, Libourne, France, **3** Attoquant Diagnostics, Vienna, Austria, **4** Department of Veterinary Biomedical Sciences, SMART Pharmacology, College of Veterinary Medicine, Iowa State University, Ames, Iowa, United States of America

¤ Current address: Department of Veterinary Clinical Sciences, College of Veterinary Medicine, University of Minnesota, St. Paul, Minnesota, United States of America
* maste289@umn.edu

## Abstract

### Objective

To characterize the dose-exposure-response effect of spironolactone on biomarkers of the classical and alternative arms of the renin-angiotensin-aldosterone system (RAAS) in healthy dogs.

### Animals

Ten healthy purpose-bred Beagle dogs.

### Procedures

Study dogs were randomly allocated to 2 spironolactone dosing groups (2 mg/kg PO q24hr, 4 mg/kg PO q24hr). The dogs received 7-day courses of spironolactone followed by a 14-day washout period in a crossover (AB/BA) design. Angiotensin peptides and aldosterone were measured in serum using equilibrium analysis, and plasma canrenone and 7-α-thio-methyl spironolactone (TMS) were quantified via liquid chromatography-mass spectrometry/mass spectroscopy (LC-MS/MS). Study results were compared before and after dosing and between groups.

### Results

Following spironolactone treatment, dogs had a significant increase in serum aldosterone concentration ($P = 0.07$), with no statistical differences between dosing groups. Significant increases in angiotensin II ($P = 0.09$), angiotensin I ($P = 0.08$), angiotensin 1–5 ($P = 0.08$), and a surrogate marker for plasma renin activity ($P = 0.06$) were detected compared to baseline following spironolactone treatment during the second treatment period only.

**Data Availability Statement:** All relevant data are within the paper and its Supporting information files.

**Funding:** YES - Funding for this study was provided by Ceva Sante Animale (http://www.ceva.com/). Author EG is an employee of Ceva Sante Animale and had a role in study design and preparation of manuscript. Authors JLW and JPM have served as consultants for Ceva Sante Animale and have received reimbursement and honoraria for consulting, expert testimony, travel, and service as key opinion leaders. JWL and JPM played a role in study design, data collection and analysis, decision to publish, and preparation of manuscript. Authors AKM, OD, and LY have no relevant competing interests.

**Competing interests:** I have read the journal's policy and the authors of this manuscript have the following competing interests: author EG is an employee of Ceva Sante Animale and authors JLW and JPM have served as consultants for Ceva Sante Animale and have received reimbursement and honoraria for consulting, expert testimony, travel, and service as key opinion leaders. Ceva Sante Animale is a multinational company that performs research, develops, manufactures and supplies vaccines, pharmaceutical medicines and other animal health products, together with the equipment, training, technical support and specialized services to ensure their optimal use. Ceva Sante Animale provided funding for this research. This does not alter our adherence to PLOS ONE policies on sharing data and materials.

**Abbreviations:** AA2, adrenal responsiveness; ABT, aldosterone breakthrough; ACE-I, angiotensin converting enzyme inhibitor; ACE-S, angiotensin converting enzyme activity; ACVIM, American college of veterinary internal medicine; ALT-S, alternative renin-angiotensin-aldosterone system activity; Ang, angiotensin; ARB, angiotensin receptor blocker; CBC, complete blood count; CHF, congestive heart failure; DCM, dilated cardiomyopathy; LC-MS/MS, liquid chromatography-mass spectrometry/mass spectroscopy; MMVD, myxomatous mitral valve disease; MRA, mineralocorticoid receptor antagonist; PRA-S, plasma renin activity; RAAS, renin-angiotensin-aldosterone system; SAP, systolic arterial blood pressure; TMS, 7-α-thiomethyl spironolactone.

Overall, changes from baseline did not significantly differ between spironolactone dosages. RAAS analytes were weakly correlated (R < 0.4) with spironolactone dosage and plasma canrenone or plasma TMS. There were no adverse clinical or biochemical effects seen at any spironolactone dosage during treatment.

## Conclusions

Treatment with spironolactone increased serum aldosterone concentration in healthy dogs and impacted other biomarkers of the classical and alternative arms of the RAAS. There was no difference in effect on the RAAS between 2 and 4 mg/kg/day dosing. Dosage of 4 mg/kg/day was safe and well-tolerated in healthy dogs.

## Introduction

The use of spironolactone, a mineralocorticoid receptor antagonist (MRA), has been associated with a significant reduction in risk of cardiac morbidity and mortality in humans [1] and dogs [2, 3] with congestive heart failure (CHF). The unique importance of MRAs in CHF management is thought to be related to mitigation of aldosterone breakthrough (ABT), [4] a phenomenon wherein individuals treated with renin-angiotensin-aldosterone system (RAAS) modulating drugs, such as angiotensin converting enzyme inhibitors (ACE-I) or angiotensin receptor blockers (ARB), exhibit an increase in plasma aldosterone concentration following treatment [5]. The exact cause of ABT is unknown. However, many mechanisms have been proposed, including: insufficient inhibition of ACE activity with ACE-I therapy, synthesis of angiotensin II (AngII) through non-ACE pathways, and genetic mutations resulting in increased circulating levels of ACE [6]. Veterinary studies have demonstrated ABT that is independent of ACE-I dose following ACE-I treatment in healthy dogs with furosemide-induced RAAS activation [7–9]. ABT has also been documented in dogs with various stages of naturally occurring myxomatous mitral valve disease (MMVD) [10] and proteinuric chronic kidney disease [11].

Due to the documented survival benefit of MRA treatment in patients with CHF and the recognized importance of ABT in cardiovascular disease pathology, the most recent consensus guidelines from the American College of Veterinary Internal Medicine (ACVIM) recommend treatment with spironolactone at a dosage of 2.0 mg/kg by mouth every 12–24 hours for aldosterone antagonism in stage C MMVD [12]. However, despite the demonstrated clinical benefit of spironolactone in canine CHF [2, 3, 13, 14], it remains unknown exactly how spironolactone affects the classical and alternative arms of the RAAS pathway. The classical arm of the RAAS pathway includes the hormones angiotensin I (AngI), AngII, and aldosterone, as well as the enzymes responsible for generation of these hormones (renin and ACE). Upregulation of the classical arm of the RAAS pathway leads to sodium and water retention, vasoconstriction, and pathologic remodeling of the myocardium. The alternative arm of the RAAS pathway includes the hormones angiotensin 1–7 (Ang1-7) and angiotensin 1–5 (Ang1-5) as well as the enzyme ACE2 which hydrolyzes AngII into Ang1-7. The alternative arm of the RAAS pathway serves as a counter-regulatory arm of the RAAS. Upregulation of the alternative arm of the RAAS leads to natriuresis and diuresis, vasodilation, anti-inflammatory and anti-proliferative effects (Fig 1) [15].

In addition to questions about impact on classical and alternative RAAS, information about the ideal dosing of spironolactone in dogs with heart disease is currently unavailable. The

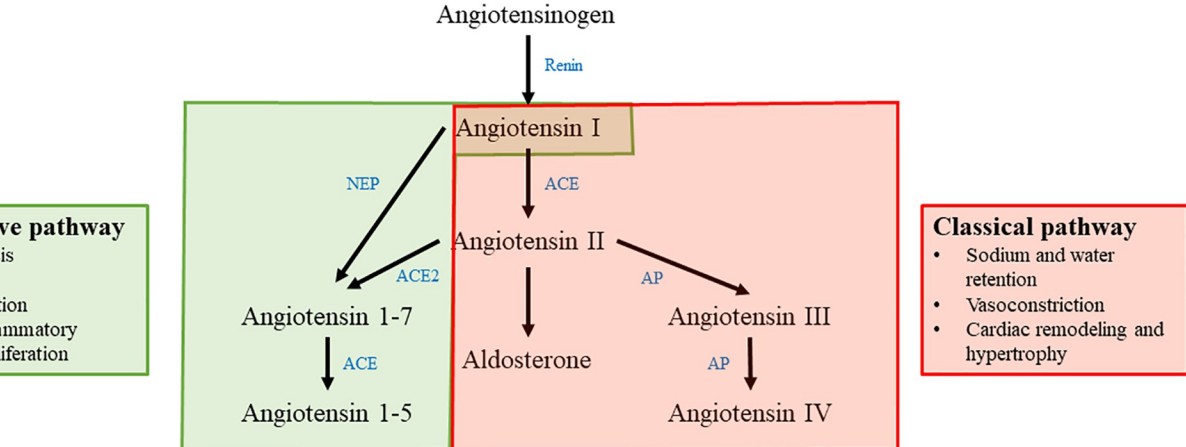

**Fig 1. Simplified depiction of the classical and alternative RAAS pathways.** The classical pathway is highlighted in red and consists of angiotensin I, angiotensin II, and aldosterone. The effects of the classical RAAS pathway are listed on the right hand side of the figure. The alternative RAAS pathway is highlighted in green and consists of angiotensin I, angiotensin 1–7, and angiotensin 1–5. The effects of the alternative RAAS pathway are listed on the left hand side of the figure. Enzymes involved in the RAAS pathway are shown in blue. NEP: neprilysin; ACE: angiotensin converting enzyme; ACE2: angiotensin converting enzyme 2; AP: aminopeptidase.

dosage of spironolactone used in previous clinical trials demonstrating survival benefit in dogs with CHF was 2–4 mg/kg PO q24hr [2, 3]. In another study, treatment with low-dosage spironolactone (0.52 mg/kg PO q24hr) did not result in any survival benefit in dogs with CHF secondary to MMVD or dilated cardiomyopathy (DCM) [16]. The purpose of the present study was to characterize the effect of different doses of spironolactone on a comprehensive array of RAAS metabolites in healthy dogs. We hypothesized that spironolactone would alter the complete RAAS profile in healthy dogs in a dose-dependent manner.

## Materials and methods

### Animals

Study subjects were ten systemically healthy, purpose-bred Beagle dogs. Dogs were 5 spayed females and 5 neutered males, 3 years of age, weighing between 7.5–12 kg. Dogs were previously deemed healthy based on physical examination by a veterinarian, routine laboratory screening (complete blood count, serum biochemical analysis), and echocardiography. The study design was approved by the Iowa State University Institutional Animal Care and Use Committee (IACUC protocol nr. 20–153).

### Study design

This prospective study followed a complete cross-over (AB/BA) two-arm design. Study dogs were randomly allocated to 2 dosing groups (spironolactone[a] at 2 mg/kg or 4 mg/kg, by mouth every 24 hours). Treatment allocation was performed at random in R version 4.2.1 using the package "psych" and the function block.random. Each treatment was given for seven consecutive days with a 14-day washout period between treatment periods. Duration of treatment and washout periods were determined by established pharmacokinetics of spironolactone in dogs, suggesting an elimination half-life of 26 hours, such that steady state would be achieved within a week of treatment [17]. Dogs were examined and blood samples were collected at baseline (prior to initiation of spironolactone administration) on Days 0 and 21 (D0 and D21), and

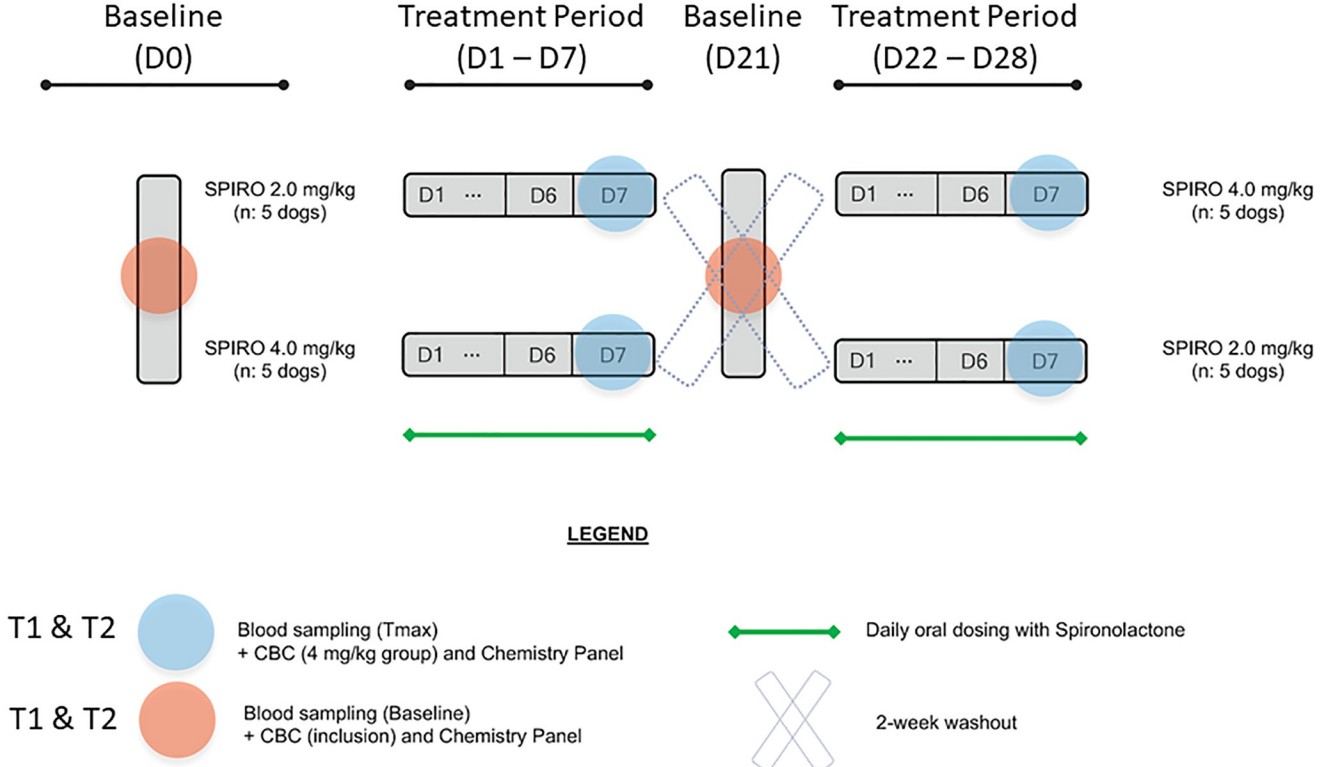

**Fig 2. Depiction of study design using a complete cross-over (AB/BA) two-arm design.** Study dogs were randomly allocated to two dosing groups (spironolactone at 2 mg/kg or 4 mg/kg, by mouth every 24 hours). Each treatment was given for seven consecutive days (D1 –D7 and D22 –D28) with a 14-day washout period between treatment periods. Samples were obtained at baseline (D0 and D21) prior to each treatment period and upon the completion of each treatment period (D7 and D28) at two timepoints: T1, immediately prior to daily spironolactone administration (07:00) and T2, peak plasma concentration (12:00).

post-treatment on Days 7 and 28 (D7 and D28). Samples were collected at two timepoints on each day, just prior to spironolactone administration (T1, 07:00) and 5 hours after receiving the assigned treatment (T2, 12:00) in order to assess differences in RAAS analytes immediately prior to daily spironolactone administration (T1) and at peak plasma concentration (T2). Spironolactone was administered at 07:00 daily on days D1 –D7 and D22 –D28 (Fig 2).

Dogs were fed their routine diet (Royal Canin Beagle diet) once daily at approximately 09:00 after treatment administration. Dogs were pair-housed in the Laboratory Animal Resources unit at Iowa State University College of Veterinary Medicine with standardized housing conditions, including a 12-hour light cycle (06:00 to 18:00) and access to water ad libitum.

On collection days (D0, D7, D21, and D28), the following data were measured at T1: body weight, heart rate, respiratory rate, serum biochemistry profile, spironolactone active metabolite (plasma canrenone and 7-$\alpha$-thiomethyl spironolactone (TMS)) concentration, and serum RAAS metabolites. A complete blood count (CBC) was also performed at T1 on D0 for all dogs and post-treatment for the treatment period when dogs received spironolactone 4 mg/kg/day (D14 or D28).

On D7 and D28, the following tests were performed at T2: plasma canrenone and TMS concentrations and serum RAAS metabolite quantification. Systolic arterial blood pressure (SAP) was measured on all collection days after T2 at 13:00.

## Procedures

Dogs were brought from their living quarters into a dedicated procedure room for data collection. Dogs were evaluated in the same order for data collection on each day and at each time point. Dogs were weighed at each time point using the same digital scale. Venous blood samples were collected from the external jugular vein using 1-inch, 20-gauge needles attached to 12 mL syringes. T1 blood samples were aliquoted into two 5 mL additive-free tubes, one chilled 5 mL lithium-heparin tube, and one 3 mL EDTA tube (if a CBC was to be performed). T2 blood samples were aliquoted into one 5 mL additive-free tube and one chilled 5 mL lithium-heparin tube. Lithium-heparin tubes were kept on ice at all times during data collection. Additive-free tubes were centrifuged at room temperature at 1,500g for 30 minutes and the resulting serum was transferred into cryovials stored at -80˚C; serum from one additive-free tube at T1 was submitted for biochemical analysis. Chilled lithium-heparin tubes were centrifuged at 4˚C at 1,500g for 20 minutes and the resulting plasma was transferred into cryovials stored at -80˚C. All CBCs and serum biochemical analyses were performed by the Iowa State University Clinical Pathology Laboratory.

Systolic arterial blood pressure was measured in a quiet room with gentle restraint by a single investigator blinded to the dog's treatment group (JW). A non-invasive Doppler ultrasonic flow probe was used following standard methods [18]. Consistent cuff sizes and patient position were used for each dog. A minimum of five consistent SAP measurements were obtained and averaged.

**Pharmacokinetic analysis.** The prodrug spironolactone has a short plasma half-life ($<2$ hours) and rapidly undergoes hepatic metabolism, resulting in the formation of several primary metabolites. Two major active metabolites include the prominent dethioacetylated metabolite, canrenone, and TMS. These active metabolites have a half-life estimated at around 15–20 hours in humans [19, 20]. Batch analysis of plasma canrenone and TMS concentrations by liquid chromatography-mass spectrometry (LC-MS/MS) were performed at each time-point. An LC-MS/MS method was developed for the quantitation of TMS and canrenone in 0.025 mL of dog lithium plasma. The method utilized canrenone-d6 as internal standards. After the addition of the internal standards, the samples were processed using crash protein precipitation with acetonitrile. Organic phase was evaporated to dryness at 30˚C under nitrogen and samples were reconstituted with a mixture 0.1% formic acid in water/acetonitrile. Chromatographic separation was achieved isocratically on an Aquity UPLC C18 column (Waters) 2.1x50 mmn, 1.7 μm at 0.40 mL/min. The mobile phase contained water, acetonitrile, and formic acid (70/30/0.1, v/v/v/). Detection was accomplished using a Sciex TQ6500+ tandem mass spectrometer in positive ion electrospray SRM mode (canrenone 341>107, TMS 389>341, and canrenone-d6 347>107). The standard curves, which ranged from 2 to 500 ng/mL, were fitted to a $1/x^2$ weighted linear regression model. The intra-assay precisions, based on three levels of QC samples (low, medium and high), were within 4.62% CV and inter-assay precisions were within 3.88% CV.

**RAAS biomarker analysis.** Batched serum samples from T1 and T2 at D0, D7, D21, and D28 were shipped frozen on dry ice to a commercial diagnostic laboratory for RAS-Fingerprint™ analysis.[b] RAAS hormones quantified in the assay include: angiotensin I (AngI), angiotensin II (AngII), angiotensin III (AngIII), angiotensin IV (AngIV), angiotensin 1–7 (Ang1-7), angiotensin 1–5 (Ang1-5), and aldosterone. Circulating RAAS analytes were quantified via LC-MS/MS at a commercial laboratory (Attoquant Diagnostics, Vienna, Austria), using previously validated and described methods [21–24]. Briefly, the assay was performed using equilibrium dialysis from serum samples that did not contain a protease inhibitor. The equilibrated serum samples were stabilized (*ex vivo* incubation at 37˚C for one hour) and spiked with stable

isotope labeled internal standards for each angiotensin metabolite as well as with the deuterated internal standard for aldosterone (aldosterone D4) at a concentration of 200pg/mL. The samples then underwent C-18-based solid-phase-extraction and were subjected to LC-MS/MS analysis using a reversed-phase analytical column (Acquity UPLC C18, Waters) operating in line with a Xevo TQ-S triple quadrupole mass spectrometer (Waters Xevo TQ/S, Milford, MA) in multiple reaction monitoring mode. Internal standards were used to correct for analyte recovery across the sample preparation procedure in each individual sample. Analyte concentrations were reported in pM and are calculated considering the corresponding response factors determined in appropriate calibration curves in sample matrix, when integrated signals exceeded a signal-to-noise ratio of 10. The lower limit of quantification was 3.0 pM for AngI, 2.0 pM for AngII, 3.0 pM for Ang1-7, 2.0 pM for Ang1-5, 2.5 pM for AngIII, 2.0 pM for AngIV, and 15 pM for aldosterone.

The following surrogate markers were calculated based on results of the RAS-Fingerprint™ analysis: plasma renin activity (PRA-S), ACE activity (ACE-S), adrenal responsiveness (AA2-r-ratio), and alternative RAAS activity (ALT-S). PRA-S, an index of plasma renin activity, was calculated as the sum of AngI + AngII. ACE-S, a measure of ACE enzyme activity, was calculated by dividing AngII / AngI. AA2, a measure of adrenal responsiveness to angiotensin II, was calculated as ALD / AngII. The ALT-S, a ratio indicating relative activity of alternative RAAS pathways compared to the classical RAAS axis, was calculated as [Ang1-7 + Ang1-5] / (AngI + AngII). All RAAS analytes and PRA-S are in pmol/L. AA2 and ACE-S are ratios of pmol/L:pmol/L.

### Statistical analysis

Possible carryover effects were evaluated by paired *t*-test between Day 0 and Day 21. The normality of each RAS-Fingerprint™ analyte was tested using Shapiro-Wilk statistics, and pairwise *t*-tests or Wilcoxon rank sum tests were performed accordingly to compare (1) percent change from baseline across timepoints between 2 vs. 4 mg/kg/day of spironolactone; (2) baseline vs. post-treatment RAS-Fingerprint™ metabolites. The coefficient of determination between the concentration of each RAS-Fingerprint™ biomarker and spironolactone active metabolites (plasma canrenone and TMS) was further calculated using linear regression models. *P*-values $< 0.1$ were considered as statistically significant. Statistical analyses were performed using commercially available software (R version 4.2.1, R Foundation for Statistical Computing, Vienna, Austria).

### Results

Samples were obtained from all dogs (n = 10) at all timepoints. The baseline (D0) CBC from one dog had clotted and therefore the baseline CBC for this dog was performed on D21. Results from the RAS-Fingerprint™, CBC, and serum biochemistry panel were not normally distributed.

### Pharmacokinetic analysis

Plasma canrenone and TMS concentrations were below the limit of detection for all dogs at baseline. Dosing with spironolactone led to dose-related increases in plasma canrenone concentrations at T1 (1.45 fold difference) and T2 (1.71 fold difference) as well as plasma TMS concentrations at T1 (1.65 fold difference) and T2 (1.97 fold difference) (Table 1).

**Table 1. Mean (one standard deviation) plasma canrenone and 7-α-thiomethyl spironolactone (TMS) concentration following treatment with spironolactone at either 2 mg/kg/day or 4mg/kg/day for seven days at T1 (immediately prior to spironolactone dosing) and T2 (5 hours post-spironolactone dose) in 10 healthy purpose-bred Beagle dogs using a cross-over study design.** The range of plasma canrenone and TMS at each timepoint following spironolactone treatment is also reported.

| Spironolactone Metabolite | 2 mg/kg/day D7 @ T1 | 2 mg/kg/day D7 @ T2 | 2 mg/kg/day D28 @ T1 | 2 mg/kg/day D28 @ T2 |
|---|---|---|---|---|
| Canrenone (ng/mL) | 6.2 (1.9) (range: 3.3–8.0) | 42.1 (6.5) (range: 33.0–50.0) | 12.2 (4.1) (range: 6.2–17.1) | 38.9 (5.1) (range: 32.2–45.1) |
| TMS (ng/mL) | 5.9 (2.7) (range: < 2.0–9.7) | 62.4 (11.4) (range: 43.6–74.7) | 7.8 (2.0) (range: 5.5–10.9) | 66.2 (21.3) (range: 37.0–87.5) |
| Spironolactone Metabolite | 4 mg/kg/day D7 @ T1 | 4 mg/kg/day D7 @ T2 | 4 mg/kg/day D28 @ T1 | 4 mg/kg/day D28 @ T2 |
| Canrenone (ng/mL) | 14.8 (10.5) (range: 8.8–33.4) | 76.3 (16.0) (range: 58.6–91.6) | 11.7 (5.0) (range: 5.6–19.5) | 61.8 (36.5) (range: 13.4–99.0) |
| TMS (ng/mL) | 14.2 (12.2) (range: 5.3–35.1) | 150.4 (62.2) (range: 82.1–234.0) | 8.5 (4.1) (range: 3.9–12.8) | 103.3 (44.6) (range: 77.6–170.0) |

## Effect of spironolactone dosage

Overall, spironolactone dosage (2 mg/kg/day vs. 4 mg/kg/day) did not have an effect on the percent change from baseline in any of the circulating RAAS analytes measured, regardless of treatment period or timepoint (Table 2). Because of the absence of an apparent linear dose-effect relationship at the doses studied, results from both dosages of spironolactone were combined to assess the effect of spironolactone on RAAS analytes.

## RAAS biomarker analysis at T1

When treatment periods and spironolactone dosages were combined, spironolactone led to an increase in all RAAS biomarkers (Fig 3), although only changes in aldosterone reached the level of statistical significance following spironolactone treatment (Table 3). However, when considering treatment periods individually, several RAAS metabolites showed significant differences from baseline during the second treatment period (D21 –D28) but not the first treatment period (D0 –D7). Specifically, there were statistically significant increases in AngII,

**Table 2. Effect of spironolactone dosage (2 mg/kg/day vs. 4 mg/kg/day) on RAS-Fingerprint™ analytes in 10 healthy purpose-bred Beagle dogs at combined baseline (D0 T1, D0 T2, D21 T1, and D21 T2) and post-treatment (D7 T1, D7 T2, D28 T1, and D28 T2) using a cross-over study design.** Data are presented as median (IQR) in pM/L for AngI, AngII, aldosterone, Ang1-7, Ang1-5, AngIII, AngIV, and PRA-S and as a ratio for ACE-S, AA2, and ALT-S. P value represents comparison of percent change from baseline between the two dosing groups.

| RAS-Fingerprint™ analyte | Baseline | Post-treatment 2 mg/kg | Post-treatment 4 mg/kg | P-value for percent change from baseline between dose groups |
|---|---|---|---|---|
| AngI(1–10) | 80.3 (50.6–127.8) | 88.2 (66.2–129.8) | 88.3 (61.5–119.0) | 0.76 |
| AngII(1–8) | 37.0 (27.5–54.6) | 47.7 (32.0–63.8) | 41.3 (33.6–60.5) | 0.76 |
| Aldosterone | 15.9 (8.7–41.2) | 34.1 (11.5–83.0) | 34.3 (16.3–52.6) | 0.62 |
| Ang1-7 | 18.0 (13.5–34.6) | 24.3 (18.4–34.5) | 25.9 (17.5–32.6) | 0.24 |
| Ang1-5 | 23.6 (16.4–42.5) | 38.3 (22.8–53.0) | 36.3 (22.1–55.5) | 0.74 |
| AngIII(2–8) | 5.5 (4.1–10.5) | 6.2 (4.6–11.9) | 5.9 (4.2–10.6) | 0.64 |
| AngIV(3–8) | 10.3 (6.7–18.1) | 10.3 (8.0–19.8) | 11.4 (8.1–16.2) | 0.82 |
| PRA-S | 119.4 (82.9–177.9) | 139.4 (97.6–192.4) | 137.9 (92.6–174.8) | 0.76 |
| ACE-S | 0.5 (0.4–0.6) | 0.5 (0.5–0.6) | 0.5 (0.5–0.6) | 0.9 |
| AA2 | 0.4 (0.2–1.0) | 0.8 (0.3–1.3) | 0.7 (0.3–1.2) | 0.62 |
| ALT-S | 0.3 (0.29–0.31) | 0.3 (0.2–0.3) | 0.3 (0.3–0.4) | 0.72 |

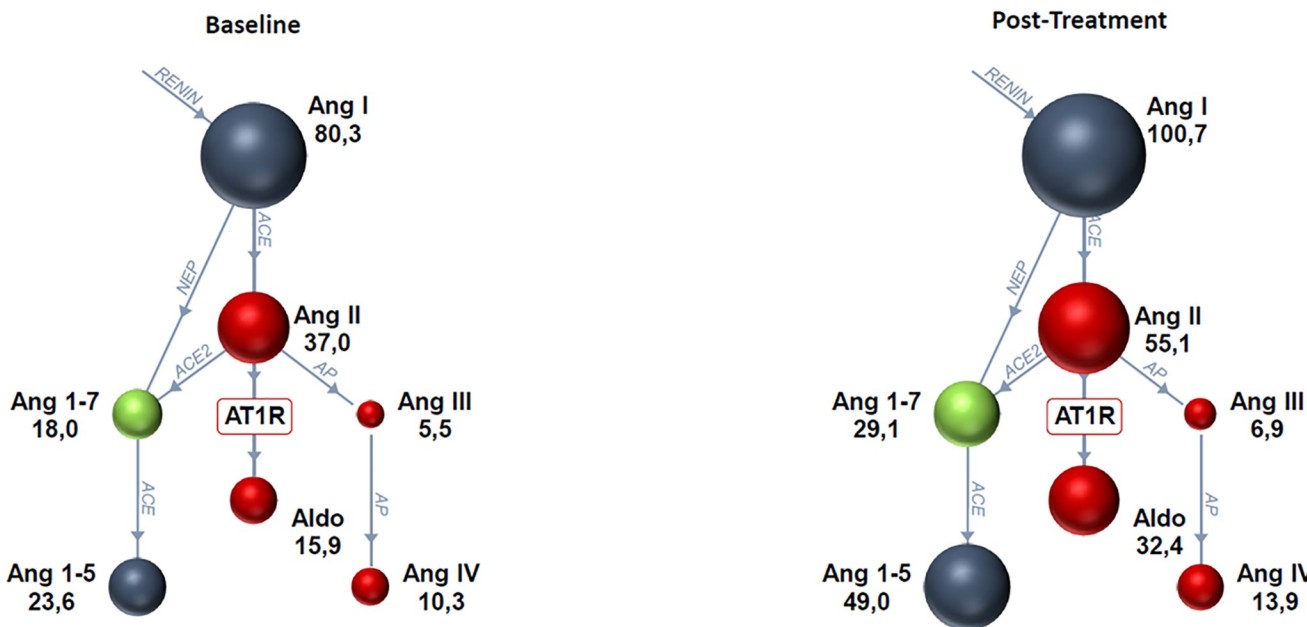

**Fig 3. RAS-Fingerprint™ analytes at baseline (combined D0 and D21) and post-treatment (D7 and D28 combined) with spironolactone at 2 mg/kg/day and 4 mg/kg/day combined at 07:00 (T1) in 10 healthy purpose-bred Beagle dogs.** There was a significant increase in aldosterone ($P < 0.1$; Table 3). Spheres show relative concentrations of angiotensin peptides. Blue spheres generally signify that the angiotensin peptide's action is inert, red indicates predominantly vasoconstrictive and pro-fibrotic effects, and green indicates vasodilatory and anti-fibrotic actions. Enzymes are shown as blue connecting lines between peptides.

AngI, Ang1-5, and PRA-S between D21 and D28 at T1 (Fig 4). These differences were not significant between D0 and D7 at T1 (Table 4).

## RAAS biomarker analysis at T2

Similar to T1, when spironolactone dosages were combined, there were changes from baseline in RAAS metabolites at T2 that were significant for the second treatment period (D21 –D28),

**Table 3. Effect of spironolactone dosage (combined 2 mg/kg/day and 4 mg/kg/day) on RAS-Fingerprint™ analytes in 10 healthy purpose-bred Beagle dogs at baseline (combined D0 and D21) and post-treatment (combined D7 and D28) using a cross-over study design at 07:00 (T1) prior to morning dosing.** Data are presented as median (IQR) in pM/L for AngI, AngII, aldosterone, Ang1-7, Ang1-5, AngIII, AngIV, and PRA-S and as a ratio for ACE-S, AA2, and ALT-S. $P$ value compares overall treatment to baseline.

| RAS Fingerprint™ analyte | Baseline | Post-treatment | Fold Difference | *P*-value for baseline vs. treatment |
|---|---|---|---|---|
| **AngI(1–10)** | 80.3 (50.6–127.8) | 100.7 (66.4–152.3) | 1.25 | 0.35 |
| **AngII(1–8)** | 37.0 (27.5–54.6) | 55.1 (33.1–79.9) | 1.49 | 0.29 |
| **Aldosterone** | 15.9 (8.7–41.2) | 32.4 (11.1–51.7) | **2.04** | **0.07** |
| **Ang1-7** | 18.0 (13.5–34.6) | 29.1 (20.2–39.1) | 1.61 | 0.34 |
| **Ang1-5** | 23.6 (16.4–42.5) | 49 (30.8–59.4) | 2.08 | 0.16 |
| **AngIII(2–8)** | 5.5 (4.1–10.5) | 6.9 (4.3–11.7) | 1.25 | 0.27 |
| **AngIV(3–8)** | 10.3 (6.7–18.1) | 13.9 (8.0–23.9) | 1.35 | 0.26 |
| **PRA-S** | 119.4 (82.9–177.9) | 154.2 (98.5–233.2) | 1.29 | 0.29 |
| **ACE-S** | 0.5 (0.4–0.6) | 0.5 (0.5–0.6) | 0 | 0.94 |
| **AA2** | 0.4 (0.2–1.0) | 0.6 (0.2–1.0) | 1.5 | 0.38 |
| **ALT-S** | 0.3 (0.3–0.3) | 0.3 (0.3–0.4) | 0 | 0.24 |

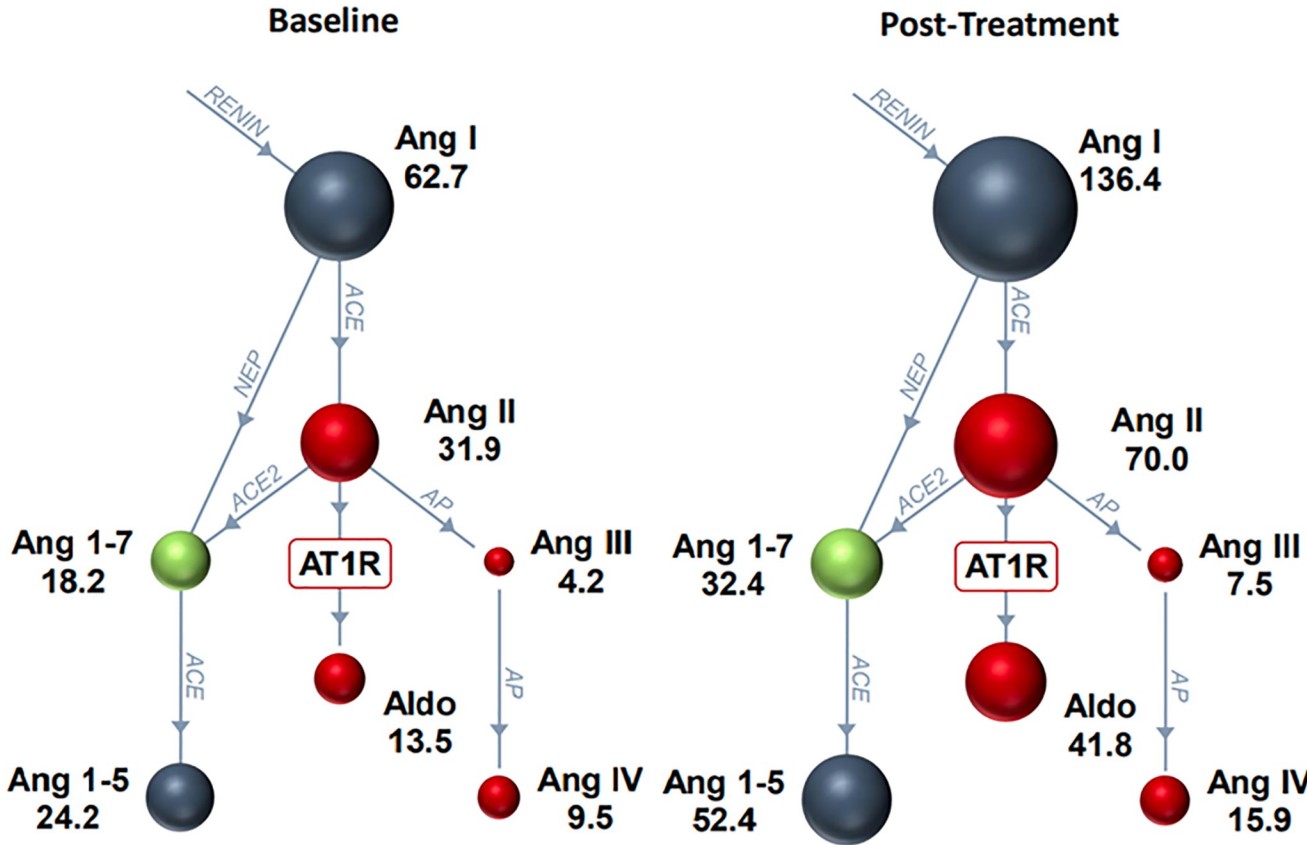

**Fig 4. RAS-Fingerprint™ analytes on Day 21 of the study (baseline) vs. Day 28 of the study (post-treatment with spironolactone at 2 mg/kg/day and 4 mg/kg/day combined) at 07:00 (T1) in 10 healthy purpose-bred Beagle dogs.** There were significant increases in AngII, AngI, Ang1-5, and PRA-S ($P < 0.1$; Table 4). Spheres show relative concentrations of angiotensin peptides. Blue spheres generally signify that the angiotensin peptide's action is inert, red indicates predominantly vasoconstrictive and pro-fibrotic effects, and green indicates vasodilatory and anti-fibrotic actions. Enzymes are shown as blue connecting lines between peptides.

**Table 4. Effect of spironolactone dosage (combined 2 mg/kg/day and 4 mg/kg/day) on RAS-Fingerprint™ analytes in 10 healthy purpose-bred Beagle dogs using a cross-over study design at each sampling period at 07:00 (T1) immediately prior to oral dosing of spironolactone.** The sampling periods include two baseline sampling periods at Day 0 (D0) and Day 21 (D21) as well as following two seven-day treatment periods at Day 7 (D7) and Day 28 (D28). Data are presented as median (IQR) in pM/L for AngI, AngII, aldosterone, Ang1-7, Ang1-5, AngIII, AngIV, and PRA-S and as a ratio for ACE-S, AA2, and ALT-S. $P$ value compares overall treatment to baseline at each specific timepoint (D0 vs. D7 and D21 vs. D28).

| RAS Fingerprint™ analyte | D0 | D7 | Fold Difference | P-value | D21 | D28 | Fold Difference | P-value |
|---|---|---|---|---|---|---|---|---|
| AngI(1–10) | 95.9 (53.1–164.9) | 81.1 (64.0–109.7) | -1.18 | 0.68 | 62.7 (38.2–112.5) | 136.4 (76.6–167.2) | **2.18** | **0.08** |
| AngII(1–8) | 45.3 (33.2–73.6) | 47.1 (29.4–59.7) | 1.04 | 0.58 | 31.9 (22.6–51.9) | 70.0 (39.4–81.2) | **2.19** | **0.09** |
| Aldosterone | 13.7 (7.0–22.0) | 32.4 (13.7–44.7) | 2.37 | 0.28 | 13.5 (9.5–36.3) | 29.4 (10.8–73.6) | 2.18 | 0.25 |
| Ang1-7 | 22.7 (14.0–44.9) | 26.1 (21.0–32.6) | 1.15 | 0.91 | 18.2 (9.6–28.0) | 32.4 (19.4–39.5) | 1.78 | 0.14 |
| Ang1-5 | 28.3 (18.0–72.2) | 41.6 (24.3–54.4) | 1.47 | 0.91 | 24.2 (16.7–40.2) | 52.4 (41.4–62.2) | **2.17** | **0.08** |
| AngIII(2–8) | 5.7 (4.3–9.5) | 6.9 (4.4–10.4) | 1.21 | 0.97 | 4.2 (2.5–6.4) | 7.5 (4.5–15.0) | 1.79 | 0.22 |
| AngIV(3–8) | 10.4 (6.5–16.9) | 12.0 (7.1–16.3) | 1.15 | 1 | 9.6 (4.8–15.8) | 15.9 (8.8–27.6) | 1.66 | 0.19 |
| PRA-S | 141.2 (84.5–236.9) | 124.4 (95.3–166.9) | -1.14 | 0.74 | 93.1 (63.2–163.7) | 207.3 (113.1–257.7) | **2.23** | **0.06** |
| ACE-S | 0.5(0.5–0.6) | 0.5 (0.5–0.6) | 0 | 0.97 | 0.6 (0.5–0.7) | 0.5 (0.5–0.7) | -1.2 | 0.85 |
| AA2 | 0.3 (0.1–0.6) | 0.7 (0.4–0.7) | 2.33 | 0.22 | 0.4 (0.3–0.6) | 0.3 (0.2–1.0) | -1.33 | 0.68 |
| ALT-S | 0.3 (0.3–0.3) | 0.3 (0.3–0.4) | 0 | 0.28 | 0.3 (0.3–0.4) | 0.3 (0.3–0.4) | 0 | 0.53 |

**Table 5. Effect of spironolactone dosage (combined 2 mg/kg/day and 4 mg/kg/day) in 10 healthy purpose-bred Beagle dogs using a cross-over study design on RAS-Fingerprint™ analytes at each sampling period at 12:00 (T2) 5-hours after oral dosing of spironolactone.** The sampling periods include two baseline sampling periods at Day 0 (D0) and Day 21 (D21) as well as following two seven-day treatment periods at Day 7 (D7) and Day 28 (D28). Data are presented as median (IQR) in pM/L for AngI, AngII, aldosterone, Ang1-7, Ang1-5, AngIII, AngIV, and PRA-S and as a ratio for ACE-S, AA2, and ALT-S. *P* value compares overall treatment to baseline at each specific timepoint (D0 vs. D7 and D21 vs. D28).

| RAS Fingerprint™ analyte | D0 | D7 | Fold Difference | *P*-value | D21 | D28 | Fold Difference | *P*-value |
|---|---|---|---|---|---|---|---|---|
| AngI(1–10) | 100.3 (60.3–121.3) | 99.4 (633.7–125.0) | 1.01 | 0.8 | 78.6 (61.0–120.0) | 79.8 (59.5–97.8) | 1.02 | 0.74 |
| AngII(1–8) | 37.0 (26.1–44.8) | 38.4 (33.7–54.8) | 1.04 | 0.68 | 34.2 (27.1–52.5) | 38.6 (31.7–55.7) | 1.13 | 0.58 |
| Aldosterone | 50.8 (19.8–68.2) | 34.2 (29.0–74.4) | -1.49 | 0.97 | 11.2 (8.8–15.7) | 33.5 (14.5–82.1) | 2.99 | 0.11 |
| Ang1-7 | 16.4 (9.5–21.4) | 21.8 (11.6–30.1) | 1.33 | 0.48 | 19.2 (14.3–33.3) | 23.2 (18.7–26.7) | 1.2 | 0.63 |
| Ang1-5 | 27.1 (16.5–37.1) | 21.9 (16.2–27.6) | -1.24 | 0.63 | 22.7 (17.7–28.0) | 35.8 (31.0–52.5) | **1.58** | **0.02** |
| AngIII(2–8) | 6.2 (4.4–10.5) | 6.1 (5.3–11.8) | -1.02 | 0.68 | 5.8 (5.2–13.1) | 5.2 (3.5–8.1) | -1.12 | 0.44 |
| AngIV(3–8) | 10.3 (9.0–16.6) | 10.8 (8.6–11.9) | 1.05 | 1 | 10.3 (8.1–20.5) | 9.5 (7.9–15.7) | -1.08 | 0.68 |
| PRA-S | 141.9 (86.2–165.4) | 137.1 (98.7–179.8) | -1.04 | 0.68 | 110.4 (88.7–172.5) | 118.4 (88.9–149.3) | 1.07 | 0.91 |
| ACE-S | 0.4 (0.4–0.5) | 0.5 (0.4–0.6) | 1.25 | 0.28 | 0.5 (0.5–0.5) | 0.5 (0.5–0.6) | 0 | 0.44 |
| AA2 | 1.4 (0.5–1.6) | 1 (0.7–1.3) | -1.4 | 0.85 | 0.4 (0.3–0.5) | 1.0 (0.5–1.4) | **2.5** | **0.06** |
| ALT-S | 0.3 (0.2–0.3) | 0.2 (0.2–0.3) | -1.5 | 0.8 | 0.3 (0.3–0.3) | 0.4 (0.3–0.4) | **1.33** | **0.002** |

but not during the first treatment period (D0 –D7). Specifically, there were significant increases in Ang1-5, AA2, and ALT-S between D21 and D28 at T2. These differences were not significant between D0 and D7 at T2 (Table 5).

## Effect of pharmacokinetics on RAAS biomarker concentration

Overall, there was wide variability in circulating RAAS analytes regardless of spironolactone dose. Visual inspection of the correlation plots between spironolactone metabolites (canrenone and TMS) and RAAS biomarkers did not suggest any correlation. Additionally, even at baseline sampling, when plasma canrenone and TMS concentrations were zero, there was wide variability in circulating RAAS analytes between dogs. As demonstrated in Fig 5, all aldosterone values but one during spironolactone treatment occurred when plasma canrenone and plasma TMS concentrations were zero. There were no significant differences between CBC values, serum biochemistry profile values, or SAP at any timepoint or spironolactone dosage. Selected biochemical and SAP data are presented in Table 6.

## Discussion

This is the first study to evaluate the effects of spironolactone treatment on the RAS-Fingerprint™ in healthy dogs. We hypothesized that mineralocorticoid receptor blockade secondary to spironolactone treatment would lead to global upregulation of the RAAS secondary to the displacement of aldosterone from its receptor leading to decreased plasma sodium concentration and thus increased renin secretion. Overall, the effects of spironolactone treatment on circulating RAAS analytes in this study were minimal and varied between study periods and as a function of time and feeding status. Previous studies have shown no effect of spironolactone treatment in healthy dogs at similar doses on water diuresis or urine excretion of sodium [25]. This contrasts with the effects of spironolactone in models of hyperaldosteronism where administration of doses as low as 1.08 mg/kg inhibited the action of aldosterone by 50% ($ED_{50}$) [26]. These findings and the results of our study suggest that in healthy dogs without background RAAS activation the physiologic effects of spironolactone are modest, and insignificant for most measures in this study, at doses known to lead to improved clinical outcomes in patients with cardiovascular pathology (2–4 mg/kg PO q24hr) [2, 3]. The modest effects of

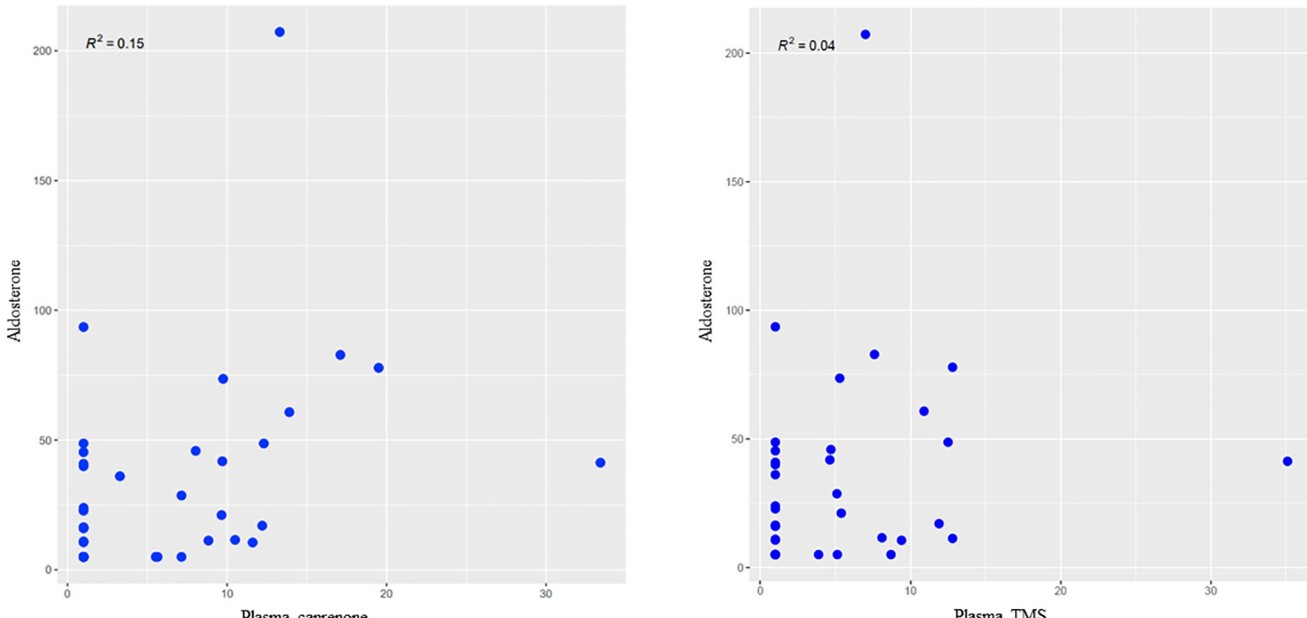

**Fig 5. Scatterplot graphs showing the weak association (R < 0.4) between plasma canrenone and plasma TMS concentration and serum aldosterone in 10 healthy purpose-bred Beagle dogs receiving spironolactone at 2mg/kg/day and 4mg/kg/day combined in a cross-over study.**

spironolactone in this study population may be explained by the small study sample size, the limited dosing range studied, and individual variability in baseline circulating RAAS. Given the clinical benefit of spironolactone treatment in clinical trials [2, 3, 13, 14], it is likely that the effect is more profound in the context of clinical disease and background RAAS activation. Therefore, future evaluation of circulating RAAS analytes in dogs with heart disease following spironolactone treatment is warranted.

Differences in spironolactone dosage of 2 mg/kg/day versus 4 mg/kg/day in this study did not lead to differential dosage effects on RAAS analytes despite proportionally increased plasma canrenone and TMS at 4 mg/kg/day vs. 2 mg/kg/day. This suggests that maximal RAAS modifying effects of spironolactone are reached by 2 mg/kg/day in the dog. This finding is consistent with the pharmacokinetic and pharmacodynamic understanding of spironolactone dosing in dogs where its effects on urinary sodium and potassium excretion reach a plateau by 2 mg/kg/day [26]. Previous studies have demonstrated the biochemical safety of spironolactone in clinical canine patients [27] and in experimental models at doses as high as

**Table 6. Effect of spironolactone treatment (combined 2 mg/kg/day and 4 mg/kg/day) in 10 healthy purpose-bred Beagle dogs using a cross-over study design on select serum biochemistry variables and SAP between baseline (combined D0 and D21) and treatment (combined D7 and D28) at 07:00 (T1; prior to feeding or morning dosing).** Data are presented as median (IQR).

| Variable | Baseline | Treatment | P-value |
|---|---|---|---|
| **SAP (mmHg)** | 133.4 (124.9–140.1) | 136.8 (124.8–141.4) | 1 |
| **Sodium (mEq/L)** | 141.5 (139.8–142.2) | 142.5 (141.0–143.0) | 0.73 |
| **Potassium (mEq/L)** | 4.2 (4.1–4.4) | 4.3 (4.2–4.3) | 0.97 |
| **BUN (mg/dL)** | 12.0 (11.0–13.0) | 11.0 (11.0–12.0) | 0.7 |
| **Creat (mg/dL)** | 0.7 (0.6–0.7) | 0.7 (0.6–0.7) | 1 |

15 mg/kg/day [28]. The development of hyperkalemia, renal dysfunction, gynecomastia in men, and gastrointestinal disturbances have been reported in humans with heart disease that are treated with spironolactone [1]. A reversible prostatic atrophy has been observed in intact male dogs treated with spironolactone [17]. The data from this study also show that spironolactone was safe in healthy dogs at doses up to 4 mg/kg/day. No dog studied experienced an adverse event while on spironolactone at either dose and spironolactone treatment had no effect on the CBC and serum biochemistry profile values or SAP in dogs in this study.

Because of the absence of a linear dose-effect relationship, the data from both the 2 mg/kg/day and 4 mg/kg/day dogs were combined for analysis of the study data presented. When analyzing combined data from both spironolactone doses at all timepoints, only serum aldosterone was found to significantly increase from baseline with spironolactone treatment. This is presumably due to displacement of aldosterone from mineralocorticoid receptors by spironolactone and subsequently increased circulating aldosterone concentrations. This is consistent with previous studies showing increased circulating serum [27] and urine [7, 13, 29] aldosterone concentrations with spironolactone treatment in dogs. In another study using low dose spironolactone at a median dose of 0.52 mg/kg/day, there was no increase in circulating plasma aldosterone. This suggests that doses as low as 0.52 mg/kg/day may be subthreshold for aldosterone displacement [16]. However, these earlier studies did not evaluate additional circulating RAAS analytes and therefore do not offer a comparison for the unexpected finding of minimal increases in other RAAS analytes with spironolactone treatment in the dogs in this study.

Historically, caution has been taken when prescribing spironolactone without additional RAAS modulating drugs such as ACE-I or ARBs in patients with heart disease due to concern for upregulation of upstream RAAS metabolites associated with negative cardiovascular effects, such as AngII. The results of this study demonstrate that beyond aldosterone, marked and significant increases in upstream RAAS components do not occur with spironolactone treatment in healthy dogs at the doses evaluated. However, in the clinical patient where global RAAS upregulation is suspected, such as dogs with CHF, [30–32] additional RAAS blockade is likely necessary. This is consistent with the current ACVIM guidelines on the treatment of stage C MMVD were combination therapy with ACE-I and spironolactone is recommended [12]. Additionally, risk of hyperkalemia is a concern in human cardiac patients treated with MRAs [1, 33]. However, no dogs in this study developed hyperkalemia with spironolactone treatment at doses up to 4 mg/kg/day. Additionally, no statistically significant change in potassium from baseline was noted in the healthy dogs in this study with spironolactone treatment. This finding is consistent with previous studies demonstrating lack of clinically significant changes in serum potassium concentration with spironolactone administration in dogs with heart disease with or without concurrent ACE-I administration [2, 3, 14, 27, 28, 34].

An unexpected finding of this study was that there were significant changes in RAAS analytes following spironolactone treatment during the second treatment period (D21–D28) that were not documented during the first treatment period (D0–D7). Specifically, between D21 and D28 we documented increases in AngII, AngI, Ang1-5, and PRA-S at T1 and increases in Ang1-5, AA2 ratio, and ALT-S at T2. These differences were not statistically significant between D0 and D7 at T1 or T2, despite having similar plasma canrenone and TMS concentrations at D7 and D28 at both T1 and T2. This may be explained by the finding that all RAAS analytes were higher at D0 compared to D21, suggesting a mild degree of global RAAS activation even at baseline for the first treatment period. In the controlled research environment of this study this finding is difficult to explain, though could be related to higher stress during the first sampling period (D0) when compared to the third (D21). Another potential reason for the lack of significant changes during the first treatment period was that the individual

variability of circulating RAAS analytes was high regardless of spironolactone dosage, and there was only weak correlation between all RAAS analytes and spironolactone dosage, peak plasma canrenone, or peak plasma TMS. Previous studies have shown large week-to-week intra-dog variability in RAAS analytes in a population of healthy dogs [15]. Additionally, even at baseline sampling, when plasma canrenone and TMS concentrations were zero, there was wide variability in RAAS analytes between dogs. Because of this, the overall variability in RAS-Fingerprint™ analytes in healthy dogs deserves further exploration with a larger sample size than the one in this study (n = 10).

Another finding when comparing RAAS analytes between T1 and T2 at D21 and D28 is the effect of feeding on the RAAS with spironolactone administration. The RAAS analytes that were statistically significantly increased with spironolactone administration at D28 compared to baseline at D21 in the fasted samples (T1) were primarily components of the classical arm of the RAAS (AngII, AngI, and PRA-S). Ang1-5, a component of the alternative arm of the RAAS, was also statistically significantly increased at T1. Conversely, the RAAS analytes that were statistically significantly increased at D28 compared to D21 after feeding (T2) were primarily components of the alternative arm of the RAAS (Ang1-5 and ALT-S). AA2, a measure of adrenal responsiveness to AngII, was also statistically significantly increased at T2. This finding is suspected to be secondary to spironolactone increasing circulating aldosterone concentration to a greater degree than circulating AngII after feeding in the dogs in this study given that previous studies have shown feeding decreases aldosterone and downregulates the RAAS [35, 36]. The effect of feeding on the RAAS in this study is confounded by the increased plasma canrenone and TMS concentrations at T2 compared to T1. The increase in spironolactone metabolites at T2 may be a reflection of peak plasma concentration of spironolactone (5-hours post-dosing) or a reflection of the effect of feeding on circulating spironolactone metabolites. The oral bioavailability of spironolactone in dogs increases to 80–90% from 50% when administered with food [26]. Future studies comparing circulating RAAS analytes in fasted dogs at peak plasma concentration of spironolactone are required to further characterize the effects on the RAAS of feeding during spironolactone administration.

The limitations of this study include the small sample size evaluated. This sample was selected based on previously performed statistical modeling using known pharmacokinetic and pharmacodynamic properties of spironolactone. However, given the individual variability in RAS-Fingerprint™ analytes in this study, larger sample sizes for future studies should be considered. The wide variability of RAAS biomarkers and small sample size in our study would be expected to increase risk of type II error; because of this, we chose to use a significance cutoff of $P < 0.1$ to focus on the extent of treatment effect to best represent the biological relevance of our results [37, 38]. However, the relationship between statistically significant changes in circulating RAAS analytes with spironolactone treatment and the biological relevance of these changes is not known. Additionally, this study evaluated circulating RAAS and therefore does not reflect the effects of spironolactone treatment on the tissue components of the RAAS. The dogs in this study were fed a commercial diet that was not sodium restricted between T1 and T2 of the study period. Therefore, the effects of spironolactone during peak plasma concentrations on the fasted RAAS cannot be determined from the data analyzed. While findings of this study demonstrate that spironolactone is operating on the plateau phase for RAAS activation at 2 mg/kg/day, lower doses of spironolactone were not tested in this study, and therefore we are unable to comment on the lowest dose of spironolactone that would maximize clinical benefit. Lastly, this study evaluated healthy, purpose-bred dogs who were expected to have no degree of background RAAS activation. The effects of spironolactone treatment on dogs with background RAAS activation likely differ from the results of this study and warrant further investigation.

## Conclusions

Spironolactone treatment at both 2 mg/kg/day and 4 mg/kg/day significantly increases serum aldosterone concentration. Consistent with previous investigations focusing on urinary sodium and potassium excretion, the effects of spironolactone on circulating RAAS metabolites reached a plateau at doses of 2 mg/kg/day, although doses up to 4 mg/kg/day were safe and well-tolerated in the healthy dogs studied. Regardless of spironolactone dosage, circulating RAAS analyte variability was high, values did not correlate with plasma canrenone or TMS concentration, and values were affected by feeding. Further evaluation of the effects of lower dosages of spironolactone on the RAAS in larger sample sizes is necessary. Additionally, evaluation of the effects of spironolactone in patients with underlying RAAS activation is warranted.

## Supporting information

**S1 Data. Excel file containing raw study data for all dogs at all timepoints (D0, D7, D21, D28).** Data includes dog identification number, spironolactone dose, average blood pressure, complete blood count and serum biochemical profile values, and RAS-Fingerprint™ analytes. **Footnotes**: ᵃSpironolactone, 25mg tablets, NorthStar Healthcare ULC, ᵇRAS-Fingerprint™, Attoquant Diagnostics, Vienna, Austria.
(XLSX)

## Author Contributions

**Conceptualization:** Emilie Guillot, Jonathan P. Mochel.

**Data curation:** Allison K. Masters, Jessica L. Ward.

**Formal analysis:** Allison K. Masters, Jessica L. Ward, Oliver Domenig, Lingnan Yuan, Jonathan P. Mochel.

**Funding acquisition:** Jessica L. Ward, Emilie Guillot, Jonathan P. Mochel.

**Investigation:** Allison K. Masters, Jessica L. Ward, Jonathan P. Mochel.

**Methodology:** Allison K. Masters, Jessica L. Ward, Emilie Guillot, Jonathan P. Mochel.

**Project administration:** Allison K. Masters, Jessica L. Ward.

**Resources:** Oliver Domenig.

**Supervision:** Jessica L. Ward.

**Visualization:** Oliver Domenig.

**Writing – original draft:** Allison K. Masters.

**Writing – review & editing:** Allison K. Masters, Jessica L. Ward, Emilie Guillot, Oliver Domenig, Lingnan Yuan, Jonathan P. Mochel.

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
