## [Decision Letter · Decision Letter 0]

1 Aug 2023

PONE-D-23-07873Comprehensive characterization of the effect of mineralocorticoid receptor antagonism with spironolactone on the renin-angiotensin-aldosterone system in healthy dogsPLOS ONE

Dear Dr. Masters,

Thank you for submitting your manuscript to PLOS ONE. After careful consideration, we feel that it has merit but does not fully meet PLOS ONE’s publication criteria as it currently stands. Therefore, we invite you to submit a revised version of the manuscript that addresses the points raised during the review process.

We look forward to receiving your revised manuscript.

Kind regards,

Doa'a G. F. Al-u'datt

Academic Editor

PLOS ONE

Journal Requirements:

"I have read the journal’s policy and the authors of this manuscript have the following competing interests: author EG is an employee of Ceva Sante Animale and authors JLW and JPM have served as consultants for Ceva Sante Animale and have received reimbursement and honoraria for consulting, expert testimony, travel, and service as key opinion leaders. Ceva Sante Animale is a multinational company that performs research, develops, manufactures and supplies vaccines, pharmaceutical medicines and other animal health products, together with the equipment, training, technical support and specialized services to ensure their optimal use. Ceva Sante Animale provided funding for this research. This does not alter our adherence to PLOS ONE policies on sharing data and materials."

3. We noted in your submission details that a portion of your manuscript may have been presented or published elsewhere. "Yes - data from this manuscript were presented at the 2022 American College of Veterinary Internal Medicine Forum in Austin, TX (June 23 - 25, 2022). Conference proceedings were published in the Journal of Veterinary Internal Medicine including an abstract presentation titled "The effect of spironolactone on the renin-angiotensin-aldosterone system in dogs" involving the initial results of the present study being submitted for publication to PLOS ONE. The full published conference proceedings were published in the Journal of Veterinary Internal Medicine titled "2022 ACVIM Forum Research Abstract Program: June 22 - October 31, 2022" (J Vet Inern Med 2022;36:2282-2454 DOI: 10.1111/jvim.16541) and are attached as a related manuscript file (authors' abstract appears on page 2299-2300). This published abstract includes limited and preliminary study data and does not include any figures or the complete study data set included in the manuscript submitted." Please clarify whether this [conference proceeding or publication] was peer-reviewed and formally published. If this work was previously peer-reviewed and published, in the cover letter please provide the reason that this work does not constitute dual publication and should be included in the current manuscript.

Reviewers' comments:

Reviewer's Responses to Questions

**Comments to the Author**

1. Is the manuscript technically sound, and do the data support the conclusions?

Reviewer #1: Yes

Reviewer #2: Partly

2. Has the statistical analysis been performed appropriately and rigorously? 

Reviewer #1: I Don't Know

Reviewer #2: Yes

3. Have the authors made all data underlying the findings in their manuscript fully available?

Reviewer #1: Yes

Reviewer #2: Yes

4. Is the manuscript presented in an intelligible fashion and written in standard English?

Reviewer #1: Yes

Reviewer #2: No

5. Review Comments to the Author

Reviewer #1: PONE-D-23-07873

Title: Comprehensive characterization of the effect of mineralocorticoid receptor antagonism

with spironolactone on the renin-angiotensin-aldosterone system in healthy dogs

General Comments:

This is an interesƟng study, using sophisƟcated techniques and analyƟcal methods not ordinarily

available to most veterinary research laboratories. Therefore, this study is valuable to report findings on

the effects that ordinarily would not have been possible. However, the high degree of variability and lack

of significance in many of the parameters measured led to findings and conclusions that are somewhat

underwhelming. Perhaps the most significant limitaƟon of this study is that the invesƟgators used

healthy Beagle dogs for their analysis. (1) Beagle dogs are known to respond to drugs differently and

have different metabolism than other dogs. (2) healthy dogs likely respond to spironolactone (and other

cardiovascular drugs) differently than dogs with heart disease. I fully agree with the authors (line 353)

that “…the results of our study suggest that in healthy dogs without background RAAS acƟvaƟon [and

heart disease] the physiologic effects of spironolactone are modest”. You should include “insignificant

for most measures in this study”.

Specific Comments:

Line 166: PharmacokineƟc analysis. You did not describe to the readers why you measured these

spironolactone metabolites and did not measure spironolactone. The readers need more informaƟon.

Is this because spironolactone is rapidly metabolized and not detected as the parent drug? Did you look

for spironolactone in plasma of treated dogs and didn’t find it? Because you already have a LCMS assay

developed, it would have been easy to include detecƟon of spironolactone in your procedure. Explain

why this was not done.

Line 172: In this secƟon the analyƟcal methods are described. The assay appears to be very complete

and adequately validated. However, the reader of the paper may need more informaƟon if they were to

duplicate this assay. It says “Chromatographic separaƟon was achieved isocraƟcally on a C18 column

2.1x50 mmn, 1.7 μm at 0.40 mL/min. The mobile phase contained water, acetonitrile and formic acid.”

Please list the source of column you used and specific packing. List the proporƟon (percent) of each

component of your mobile phase. Likewise, in the next secƟon you didn’t list the ions you monitored

(m/z) ions or ranges. Please also list the lower limit of quanƟficaƟon for your assay. You used a

signal/noise raƟo (s/n) of 5 for the LOQ. Is this standard for your lab? Seems a bit low for some

guidelines.

Line 195: It says “triple quadruple mass spectrometer “. Don’t you mean “triple quadrupole”?

Line 230: In this secƟon you listed a lot of specific results that are beƩer represented in tables. Do not

repeat results in your results text if it is already listed in tables. Just refer to the tables and make some

summary comments.

Table 1: Do not say “plus or minus” one standard deviaƟon when lisƟng results in a table. This is not

staƟsƟcally accurate. List the standard deviaƟon of your sample in parentheses next the mean value.

Line 249: This secƟon addresses “Effect of Spironolactone Dosage”. Are the results listed in table 1 dose

proporƟonal? Do the metabolites measured increase by approximately 2 fold, with the increase in dose?

As it states in line 319, there is no apparent relaƟonship. Line 358 also acknowledges the lack of dose

effect.

The discussion secƟon is quite long. It is oŌen observed that when there is a lack of significance in a

study, or if results do not agree with an author’s assumpƟons, the discussion is quite long to explain why

this may have occurred (unnecessary speculaƟon). However, your discussion can be shortened

considerably by simply acknowledging that you do not know why these results occurred. Avoid

unnecessary speculaƟon to shorten your discussion.

Overall: I think the study is worthy of publicaƟon but there are many limitaƟons. The authors have cited

these limitaƟons (small sample, normal healthy dogs, etc.). But overall, based on these results we

cannot conclude that spironolactone has a clear benefit in dogs, parƟcularly on the RAAS cascade. It is

unclear, based on this evidence, how administraƟon of spironolactone is assumed to be beneficial in

dogs with heart disease, and recommended, without quesƟon, in some protocols. Moreover, how did

the products on the market get approved by regulatory authoriƟes? Perhaps these points deserve

menƟon by the authors in their discussion.

Reviewer #2: Authors evaluate the effect of mineralocorticoid receptor antagonism with spironolactone on the renin-angiotensin-aldosterone system in healthy dogs. The study is quite interesting and useful However, I suggest following changes to improve the quality of the manuscript for readers and research community.

Comments

Comment 1: Authors mentioned several claims in introduction section regarding the association of spironolactone with reduced risk of cardiac morbidity and mortality in humans and dogs with CHF. However, there are no specific references provided for these statements. While the introduction introduces the concept of aldosterone breakthrough (ABT) in the context of ACE-I treatment, it fails to provide a comprehensive explanation of ABT and its underlying mechanisms. Furthermore, this section contains certain ambiguous statements which may require further clarification. For example, the mention of "genetic mutations in ACE" as a proposed mechanism of ABT lacks context and requires more elaboration.

Comment 2: The study used a total of ten Beagle dogs, five in each dosing group, for the complete cross-over (AB/BA) two-arm design. While the authors mentioned random allocation to dosing groups, they did not elaborate on the method used for randomization or any power analysis to determine the sample size. It would be more appropriate to provide more details on the randomization process and justify the sample size to statistical design to draw meaningful conclusions.

Comment 3: The authors mentioned that the effects of spironolactone treatment on circulating RAAS analytes in healthy dogs were minimal and varied between study periods and as a function of time and feeding status. However, the discussion does not provide a thorough explanation for the observed minimal effects. While the study provides valuable insights into spironolactone's effects on RAAS in healthy dogs, the discussion should also discuss future research directions and potential areas for further investigation.

Comment 4: Safety profile of the studied drugs i.e. spironolactone should be addressed.

Comment 5: Limitation of the study should be discussed in limitation section of the manuscript.

Comment 6: Authors should describe the Future perspective and clinical significance of the study.

Comment 7: Authors should add abbreviation list used in the manuscript.

6. PLOS authors have the option to publish the peer review history of their article (what does this mean?). If published, this will include your full peer review and any attached files.

Reviewer #1: No

Reviewer #2: **Yes: **Mehmood Ahmad

---

## [Author Response · Author response to Decision Letter 0]

13 Sep 2023

Manuscript: Comprehensive characterization of the effect of mineralocorticoid receptor antagonism with spironolactone on the renin-angiotensin-aldosterone system in healthy dogs.

Response to PLOS ONE reviewers: PONE-D-23-07873

Dear PLOS ONE reviewers,

Thank you for the constructive reviews and edits to our manuscript “Comprehensive characterization of the effect of mineralocorticoid receptor antagonism with spironolactone on the renin-angiotensin-aldosterone system in healthy dogs.” We appreciate the time and effort that you put into reviewing our manuscript and providing insightful feedback. Please see our responses below, in bold, to the comments provided.

Reviewer #1:

1. This is an interesƟng study, using sophisƟcated techniques and analyƟcal methods not ordinarily available to most veterinary research laboratories. Therefore, this study is valuable to report findings on the effects that ordinarily would not have been possible. However, the high degree of variability and lack of significance in many of the parameters measured led to findings and conclusions that are somewhat underwhelming. Perhaps the most significant limitaƟon of this study is that the invesƟgators used healthy Beagle dogs for their analysis. (1) Beagle dogs are known to respond to drugs differently and have different metabolism than other dogs. (2) healthy dogs likely respond to spironolactone (and other cardiovascular drugs) differently than dogs with heart disease. I fully agree with the authors (line 353) that “…the results of our study suggest that in healthy dogs without background RAAS acƟvaƟon [and heart disease] the physiologic effects of spironolactone are modest”. You should include “insignificant for most measures in this study”.

The modifier “and insignificant for most measures in this study” has been added to the statement previously on line 353. 

2. Line 166: PharmacokineƟc analysis. You did not describe to the readers why you measured these spironolactone metabolites and did not measure spironolactone. The readers need more informaƟon. Is this because spironolactone is rapidly metabolized and not detected as the parent drug? Did you look for spironolactone in plasma of treated dogs and didn’t find it? Because you already have a LCMS assay developed, it would have been easy to include detecƟon of spironolactone in your procedure. Explain why this was not done.

You are correct, spironolactone is a prodrug with a short plasma half-life (less than two hours). It rapidly undergoes hepatic metabolism, resulting in the formation of several primary metabolites, two of which act as a major active metabolites: 7α-thiomethyl-spironolactone (TMS) and the prominent dethioacetylated metabolite, canrenone. These active metabolites have a half-life estimated at around 15 – 20 hours in humans. Clarification regarding these metabolites was added to the manuscript under “Pharmacokinetic Analysis.”

References have been added to the manuscript and appear below:

Kolkhof P, Bärfacker L. 30 years of the mineralocorticoid receptor: mineralocorticoid receptor antagonists: 60 years of research and development. J Endocrinol. 2017 Jul;234(1):T125-T140. doi: 10.1530/JOE-16-0600. PMCID: 28634268. PMID: PMC5488394

Struthers AD, Unger T. Physiology of aldosterone and pharmacology of aldosterone blockers. Eur Heart J Supplements, Volume 13, Issue supple B, July 2011, Pages B27-B30, https://doi.org/10.1093/eurheartj/sur009.

3. Line 172: In this secƟon the analyƟcal methods are described. The assay appears to be very complete and adequately validated. However, the reader of the paper may need more informaƟon if they were to duplicate this assay. It says “Chromatographic separaƟon was achieved isocraƟcally on a C18 column 2.1x50 mmn, 1.7 μm at 0.40 mL/min. The mobile phase contained water, acetonitrile and formic acid.” Please list the source of column you used and specific packing. List the proporƟon (percent) of each component of your mobile phase. Likewise, in the next secƟon you didn’t list the ions you monitored (m/z) ions or ranges. Please also list the lower limit of quanƟficaƟon for your assay. You used a signal/noise raƟo (s/n) of 5 for the LOQ. Is this standard for your lab? Seems a bit low for some guidelines.

Additional information was provided to allow for accurate duplication of the pharmacokinetic analysis performed in this study. Specifically, the source of the column (Acquity UPLC), proportion of each component of the mobile phase (70/30/0.1), and the ions monitored (canrenone 341>107, TMS 389>341, and canrenone-d6 347>107) were added to the revised manuscript. The lower limit of detection was 2 ng/mL. The signal/noise ratio statement was removed as the specificity of the method is more appropriately described in the sentence prior which states “the intra-assay precisions, based on three levels of QC samples (low, medium and high), were within 4.62 % CV and inter-assay precisions were within 3.88 % CV.”

4. Line 195: It says “triple quadruple mass spectrometer “. Don’t you mean “triple quadrupole”?

Yes, thank you for catching this error. This has been corrected in the manuscript. 

5. Line 230: In this secƟon you listed a lot of specific results that are beƩer represented in tables. Do not repeat results in your results text if it is already listed in tables. Just refer to the tables and make some summary comments.

Text represented in Table 1 was removed from this section to avoid duplication of information. 

6. Table 1: Do not say “plus or minus” one standard deviaƟon when lisƟng results in a table. This is not staƟsƟcally accurate. List the standard deviaƟon of your sample in parentheses next the mean value. Line 249: This secƟon addresses “Effect of Spironolactone Dosage”. Are the results listed in table 1 dose proporƟonal? Do the metabolites measured increase by approximately 2 fold, with the increase in dose? As it states in line 319, there is no apparent relaƟonship. Line 358 also acknowledges the lack of dose effect.

This correction has been made to table 1 in the manuscript and is reflected in the table heading and the body of the table. The observation regarding dose proportional effect of spironolactone on TMS and canrenone concentrations is insightful. Examining each of the two study periods separately, the fold-change in estimated canrenone exposure following administration of 4 mg/kg spironolactone compared to 2 mg/kg at T1 (immediately prior to spironolactone dosing) and T2 (5 hours post-spironolactone dose) was found to be 2.4 and 1.8 respectively, for D7; and 1 (T1) and 1.6 (T2) for D28. Concerning TMS, the fold-change in estimated exposure after 4 mg/kg spironolactone dosing in comparison to 2 mg/kg at T1 and T2 was measured at 2.4 and 2.4, respectively, for D0; and 1.1 (T1) and 1.6 (T2) for D28.

Pooling data from all study periods, the fold-change in estimated canrenone exposure following a doubling of the oral spironolactone dose stood at 1.4 and 1.7 at T1 and T2, respectively. For TMS, these estimates were 1.7 (T1) and 2.0 (T2). Taken together, these findings suggest a quasi-proportional relationship between the exposure of active spironolactone metabolites and the administered dose of spironolactone. Nonetheless, it is essential to approach the conclusions of this study with caution due to the study’s inherent limitation, including a small subject pool and the utilization of a sparse sampling approach (limited to two timepoints in this instance). 

Our data suggest that the effect of spironolactone active metabolites on biomarkers of the RAAS are not dose-proportional, with a plateauing of the effect already at 2 mg/kg/day of spironolactone.

7. The discussion secƟon is quite long. It is oŌen observed that when there is a lack of significance in a study, or if results do not agree with an author’s assumpƟons, the discussion is quite long to explain why this may have occurred (unnecessary speculaƟon). However, your discussion can be shortened considerably by simply acknowledging that you do not know why these results occurred. Avoid unnecessary speculaƟon to shorten your discussion.

The authors acknowledge the length of the discussion section reflects the lack of significance in the study dataset. The discussion section was modified to remove any unnecessary speculation while also addressing reviewer #2 comment 3 requesting a more thorough explanation for the observed minimal effects of spironolactone on RAAS analytes in the study dataset. Specifically, the authors removed the statement “this may represent a true increase in adrenal responsiveness to AngII and subsequently increased aldosterone production secondary to feeding” from the manuscript’s discussion of the effects of feeding on the RAAS.

8. Overall: I think the study is worthy of publicaƟon but there are many limitaƟons. The authors have cited these limitaƟons (small sample, normal healthy dogs, etc.). But overall, based on these results we cannot conclude that spironolactone has a clear benefit in dogs, parƟcularly on the RAAS cascade. It is unclear, based on this evidence, how administraƟon of spironolactone is assumed to be beneficial in dogs with heart disease, and recommended, without quesƟon, in some protocols. Moreover, how did the products on the market get approved by regulatory authoriƟes? Perhaps these points deserve menƟon by the authors in their discussion.

The study reported in this manuscript was not designed to assess clinical benefit of spironolactone in dogs. The benefit to dogs with heart disease must be shown in dogs with disease and must evaluate clinical outcomes. This study was an initial exploratory study looking at short-term RAAS outcomes for increasing doses of spironolactone. These data can be used to optimize future clinical trials that look at clinical endpoints by showing that we do not need to use doses of spironolactone higher than 2mg/kg/day. Additionally, in the clinical patient spironolactone is typically not used as monotherapy, and its RAAS effect (e.g. preventing ABT) could be more profound in the context of concurrent ACE inhibition. The discussion section of the manuscript was modified to reflect this feedback (discussion paragraph 1; line 387 in “Revised Manuscript with Track Changes”). 

The regulatory approval of spironolactone for use in dogs was obtained prior to the availability of RAAS fingerprint analysis and therefore did not directly evaluate the effects of spironolactone on individual RAAS analytes (citation #2 and #16 in the study). At the time, the effect of different doses of spironolactone in healthy Beagle dogs on urinary sodium and potassium levels was used as a surrogate marker of degree of mineralocorticoid receptor antagonism for the registration and approval of spironolactone (citation #25 in the study). However, these surrogate markers may not be an accurate representation of the cardiovascular effects of the drug. This study is the first to provide data using the RAAS fingerprint in healthy dogs treated with spironolactone and provides a better understanding of the direct effects of spironolactone on the individual components of the RAAS than previous studies. 

Reviewer #2

1. Authors mentioned several claims in introduction section regarding the association of spironolactone with reduced risk of cardiac morbidity and mortality in humans and dogs with CHF. However, there are no specific references provided for these statements. While the introduction introduces the concept of aldosterone breakthrough (ABT) in the context of ACE-I treatment, it fails to provide a comprehensive explanation of ABT and its underlying mechanisms. Furthermore, this section contains certain ambiguous statements which may require further clarification. For example, the mention of "genetic mutations in ACE" as a proposed mechanism of ABT lacks context and requires more elaboration.

The authors cite the following studies demonstrating reduced risk of cardiac morbidity and mortality in humans and dogs with CHF:

Citation #1: Pitt B, Zannad F, Remme WL, Cody R, Castaigne A, Perez A, et al. The effect of spironolactone on morbidity and mortality in patients with severe heart failure. New Engl J Med. 2008;341: 709–717.

This study was terminated early due to an interim analysis demonstrating that spironolactone significantly reduced the risk of death by 30% among human patients with severe heart failure and a left ventricular ejection fraction < 35%. Patients in the spironolactone group also had a significant improvement in their New York Heart Association functional class. 

Citation #2: Bernay F, Bland JM, Ha J, Baduel L, Combes B, Lopex A, et al. Efficacy of spironolactone on survival in dogs with naturally occurring mitral regurgitation caused by myxomatous mitral valve disease. J Vet Intern Med 2010;24: 331–341.

This study demonstrated spironolactone treatment in dogs with myxomatous mitral valve disease significantly decreased the risk of reaching the composite endpoint (cardiac death, euthanasia because of mitral regurgitation, and worsening mitral regurgitation) by 55% (HR = 0.45). Spironolactone treatment reduced the risk of cardiac-related death or euthanasia in this study population by 69% (HR = 0.31). 

Citation #3: Coffman M, Guillot E, Blondel T, Garelli-Paar C, Feng S, Heartsill S, et al. Clinical efficacy of benazepril and spironolactone combination in dogs with congestive heart failure due to myxomatous mitral valve disease: The Benazepril Spironolactone Study (BESST). J Vet Intern Med 2021; 1–15. 

This study BESST demonstrated that treatment with combination spironolactone + benazepril in dogs significantly reduced risk of dying or worsening from cardiac causes by 27% (HR = 0.73) compared to benazepril treatment alone. 

Citation #14: Laskary A, Fonfara S, Chambers H, O’Sullivan ML. Prospective clinical trial evaluating spironolactone in Doberman pinschers with congestive heart failure due to dilated cardiomyopathy. J Vet Cardiol 2022;40: 84–98.

This study demonstrated that the development of atrial fibrillation was significantly reduced in Doberman pinscher dogs with congestive heart failure secondary to dilated cardiomyopathy receiving spironolactone treatment when compared to those receiving placebo. 

A more detailed description of aldosterone breakthrough has been added to strengthen the introduction section of the manuscript. Clarification regarding previously documented genetic mutations in ACE was also added to this section of the manuscript. 

2. The study used a total of ten Beagle dogs, five in each dosing group, for the complete cross-over (AB/BA) two-arm design. While the authors mentioned random allocation to dosing groups, they did not elaborate on the method used for randomization or any power analysis to determine the sample size. It would be more appropriate to provide more details on the randomization process and justify the sample size to statistical design to draw meaningful conclusions.

The randomization was performed in R version 4.2.1 using the package “psych” and the function block.random. Clarification regarding the randomization process was added to the revised manuscript.

3. The authors mentioned that the effects of spironolactone treatment on circulating RAAS analytes in healthy dogs were minimal and varied between study periods and as a function of time and feeding status. However, the discussion does not provide a thorough explanation for the observed minimal effects. While the study provides valuable insights into spironolactone's effects on RAAS in healthy dogs, the discussion should also discuss future research directions and potential areas for further investigation.

The discussion section was modified to include a more thorough explanation for the observed minimal effects while being mindful of reviewer #1 comment 7 suggesting avoidance of any unnecessary speculation in the manuscript discussion.

4. Safety profile of the studied drugs i.e. spironolactone should be addressed.

The safety profile of spironolactone was expanded upon in the discussion section to include mention of previously documented adverse events associated with spironolactone treatment in humans. Availability of previously documented side effects of spironolactone in dogs are limited.

5. Limitation of the study should be discussed in limitation section of the manuscript.

Study limitations are addressed in the final paragraph of the discussion section (discussion paragraph 7; line 483 in “Revised Manuscript with Track Changes”). A separate study limitations section was not created in accordance with the style of other similar publications in PLOS ONE. 

6. Authors should describe the Future perspective and clinical significance of the study.

The authors recommend future evaluation of circulating RAAS analytes in dogs with underlying RAAS activation, such as those with heart disease, following spironolactone treatment at broader dose ranges. These recommendations are detailed in the conclusion section of the manuscript. 

7. Authors should add abbreviation list used in the manuscript.

The authors have added an abbreviation list to the start of the manuscript that includes all abbreviations referenced throughout the manuscript.

---

## [Decision Letter · Decision Letter 1]

7 Nov 2023

PONE-D-23-07873R1Comprehensive characterization of the effect of mineralocorticoid receptor antagonism with spironolactone on the renin-angiotensin-aldosterone system in healthy dogsPLOS ONE

Dear Dr. Masters,

Thank you for submitting your manuscript to PLOS ONE. After careful consideration, we feel that it has merit but does not fully meet PLOS ONE’s publication criteria as it currently stands. Therefore, we invite you to submit a revised version of the manuscript that addresses the points raised during the review process.

We look forward to receiving your revised manuscript.

Kind regards,

Doa'a G. F. Al-u'datt

Academic Editor

PLOS ONE

Journal Requirements:

Reviewers' comments:

Reviewer's Responses to Questions

**Comments to the Author**

1. If the authors have adequately addressed your comments raised in a previous round of review and you feel that this manuscript is now acceptable for publication, you may indicate that here to bypass the “Comments to the Author” section, enter your conflict of interest statement in the “Confidential to Editor” section, and submit your "Accept" recommendation.

Reviewer #2: All comments have been addressed

Reviewer #3: (No Response)

2. Is the manuscript technically sound, and do the data support the conclusions?

Reviewer #2: Partly

Reviewer #3: Yes

3. Has the statistical analysis been performed appropriately and rigorously? 

Reviewer #2: Yes

Reviewer #3: I Don't Know

4. Have the authors made all data underlying the findings in their manuscript fully available?

Reviewer #2: Yes

Reviewer #3: Yes

5. Is the manuscript presented in an intelligible fashion and written in standard English?

Reviewer #2: Yes

Reviewer #3: Yes

6. Review Comments to the Author

Reviewer #2: I am pleased to recommend the acceptance of the manuscript for publication, as the authors have diligently addressed all the comments and concerns raised during the review process. The revisions made have significantly improved the quality and clarity of the article.

Reviewer #3: The manuscript entitled “Comprehensive characterization of the effect of mineralocorticoid receptor antagonism with spironolactone on the renin-angiotensin-aldosterone system in healthy dogs” describes the effects of a mineralocorticoid receptor antagonist on the plasma levels of numerous RAAS components in dogs.

The research question addressed is highly relevant for veterinary medicine with increasing numbers of dogs being affected by heart disease and treated with mineralocorticoid receptor antagonists. The manuscript is well written and the interpretation of the results is scientifically sound. The low number of dogs enrolled as a major limitation of the study is already being discussed by the authors and is a common problem in this kind of studies. Hence, conducting this study with a homogenous group of n = 10 dogs and even enhancing the outcome by the elegant crossover design must be considered an achievement.

I have only minor comments and would recommend accepting the manuscript for publication.

Minor comments:

1) L.90-99: At least one reference for all the information on the RAAS should be added.

2) L. 176: “red top tube” is not very specific, as this may vary.

3) Although there is a statistically significant difference in some values for RAAS plasma levels, the biological relevance of an increase by factor 1.04 (and generally < 2) is highly questionable. This should be stated clearly in the discussion.

4) L.480: warrant (not warrants)

7. PLOS authors have the option to publish the peer review history of their article (what does this mean?). If published, this will include your full peer review and any attached files.

Reviewer #2: **Yes: **Mehmood Ahmad

Reviewer #3: No

---

## [Author Response · Author response to Decision Letter 1]

1 Dec 2023

Manuscript: Comprehensive characterization of the effect of mineralocorticoid receptor antagonism with spironolactone on the renin-angiotensin-aldosterone system in healthy dogs

Response to PLOS ONE reviewers: PONE-D-23-07873R1

Dear PLOS ONE reviewers,

Thank you for the continued consideration of our manuscript “Comprehensive characterization of the effect of mineralocorticoid receptor antagonism with spironolactone on the renin-angiotensin-aldosterone system in healthy dogs.” We appreciate the time and effort that you put into reviewing our previous edits and continuing to provide insightful feedback. Please see our responses below, in bold, to the comments provided.

Reviewer #2:

I am pleased to recommend the acceptance of the manuscript for publication, as the authors have diligently addressed all the comments and concerns raised during the review process. The revisions made have significantly improved the quality and clarity of the article.

Thank you for taking the time to provide constructive feedback that improved the quality and clarity of the article. We are sincerely grateful for your time and expertise. 

Reviewer #3:

The manuscript entitled “Comprehensive characterization of the effect of mineralocorticoid receptor antagonism with spironolactone on the renin-angiotensin-aldosterone system in healthy dogs” describes the effects of a mineralocorticoid receptor antagonist on the plasma levels of numerous RAAS components in dogs.

The research question addressed is highly relevant for veterinary medicine with increasing numbers of dogs being affected by heart disease and treated with mineralocorticoid receptor antagonists. The manuscript is well written and the interpretation of the results is scientifically sound. The low number of dogs enrolled as a major limitation of the study is already being discussed by the authors and is a common problem in this kind of studies. Hence, conducting this study with a homogenous group of n = 10 dogs and even enhancing the outcome by the elegant crossover design must be considered an achievement.

I have only minor comments and would recommend accepting the manuscript for publication.

Minor comments:

1) L.90-99: At least one reference for all the information on the RAAS should be added.

The following citation was added to clarify and support the information on the RAAS in L90-99: 

Citation #15: Hammond HK, Ames MK, Domenig O, Scansen BA, Tsang Yang N, Wilson MD, Sunshine E, Brunk K, Masters A. The classical and alternative circulating renin-angiotensin system in normal dogs and dogs with stage B1 and B2 myxomatous mitral valve disease. J Vet Intern Med 2023;37:875–886. 

2) L. 176: “red top tube” is not very specific, as this may vary.

The language in L176 was changed to “additive-free tube” in order to be consistent with the language previously used in the methods section and to most accurately describe the tube type used. 

3) Although there is a statistically significant difference in some values for RAAS plasma levels, the biological relevance of an increase by factor 1.04 (and generally < 2) is highly questionable. This should be stated clearly in the discussion.

A comment was added in the limitations section of the paper at L470 addressing the unknown relationship between statistical relevance and biological relevance in our data given the changes from baseline where typically < 2 fold. 

4) L.480: warrant (not warrants)

This change was made in L480.

---

## [Decision Letter · Decision Letter 2]

17 Jan 2024

Comprehensive characterization of the effect of mineralocorticoid receptor antagonism with spironolactone on the renin-angiotensin-aldosterone system in healthy dogs

PONE-D-23-07873R2

Dear Dr. Masters,

We’re pleased to inform you that your manuscript has been judged scientifically suitable for publication and will be formally accepted for publication once it meets all outstanding technical requirements.

Kind regards,

Doa'a G. F. Al-u'datt

Academic Editor

PLOS ONE

Additional Editor Comments (optional):

Reviewers' comments:

Reviewer's Responses to Questions

**Comments to the Author**

1. If the authors have adequately addressed your comments raised in a previous round of review and you feel that this manuscript is now acceptable for publication, you may indicate that here to bypass the “Comments to the Author” section, enter your conflict of interest statement in the “Confidential to Editor” section, and submit your "Accept" recommendation.

Reviewer #2: All comments have been addressed

Reviewer #3: All comments have been addressed

2. Is the manuscript technically sound, and do the data support the conclusions?

Reviewer #2: Partly

Reviewer #3: Yes

3. Has the statistical analysis been performed appropriately and rigorously? 

Reviewer #2: Yes

Reviewer #3: Yes

4. Have the authors made all data underlying the findings in their manuscript fully available?

Reviewer #2: Yes

Reviewer #3: Yes

5. Is the manuscript presented in an intelligible fashion and written in standard English?

Reviewer #2: Yes

Reviewer #3: Yes

6. Review Comments to the Author

Reviewer #2: Authors have diligently addressed almost all the comments and concerns raised during the review process. The revisions made have significantly improved the quality and clarity of the article.

Reviewer #3: (No Response)

7. PLOS authors have the option to publish the peer review history of their article (what does this mean?). If published, this will include your full peer review and any attached files.

Reviewer #2: **Yes: **Mehmood Ahmad

Reviewer #3: No

---

## [Editor Report · Acceptance letter]

13 Feb 2024

PONE-D-23-07873R2 

PLOS ONE

Dear Dr. Masters, 

I'm pleased to inform you that your manuscript has been deemed suitable for publication in PLOS ONE. Congratulations! Your manuscript is now being handed over to our production team.

Kind regards, 

on behalf of

Dr. Doa'a G. F. Al-u'datt 

Academic Editor

PLOS ONE